# Co-infection of the four major *Plasmodium* species: Effects on densities and gametocyte carriage

**Aurel Holzschuh**[1,2,¤], **Maria Gruenberg**[1,2], **Natalie E. Hofmann**[1,2], **Rahel Wampfler**[1,2], **Benson Kiniboro**[3], **Leanne J. Robinson**[3,4,5], **Ivo Mueller**[4,5], **Ingrid Felger**[1,2]*, **Michael T. White**[6]*

**1** Swiss Tropical and Public Health Institute, Basel, Switzerland, **2** University of Basel, Basel, Switzerland, **3** Vector Borne Diseases Unit, Papua New Guinea Institute of Medical Research, Madang and Maprik, Papua New Guinea, **4** Walter and Eliza Hall Institute of Medical Research, Melbourne, Australia, **5** University of Melbourne, Melbourne, Australia, **6** Institut Pasteur, Université de Paris Cité, G5 Épidémiologie et Analyse des Maladies Infectieuses, Département de Santé Globale, Paris, France

¤ Current address: Department of Biological Sciences, University of Notre Dame, Notre Dame, Indiana, United States of America
* ingrid.felger@unibas.ch (IF); michael.white@pasteur.fr (MTW)

**Data Availability Statement:** All data and code used for producing the results are freely available on GitHub: https://github.com/MWhite-InstitutPasteur/malaria_coinfection_PNG.

## Abstract

### Background

Co-infection of the four major species of human malaria parasite *Plasmodium falciparum* (*Pf*), *P. vivax* (*Pv*), *P. malariae* (*Pm*), and *P. ovale* sp. (*Po*) is regularly observed, but there is limited understanding of between-species interactions. In particular, little is known about the effects of multiple *Plasmodium* species co-infections on gametocyte production.

### Methods

We developed molecular assays for detecting asexual and gametocyte stages of *Pf*, *Pv*, *Pm*, and *Po*. This is the first description of molecular diagnostics for *Pm* and *Po* gametocytes. These assays were implemented in a unique epidemiological setting in Papua New Guinea with sympatric transmission of all four *Plasmodium* species permitting a comprehensive investigation of species interactions.

### Findings

The observed frequency of *Pf-Pv* co-infection for asexual parasites (14.7%) was higher than expected from individual prevalence rates (23.8%*Pf* x 47.4%*Pv* = 11.3%). The observed frequency of co-infection with *Pf* and *Pv* gametocytes (4.6%) was higher than expected from individual prevalence rates (13.1%*Pf* x 28.2%*Pv* = 3.7%). The excess risk of co-infection was 1.38 (95% confidence interval (CI): 1.09, 1.67) for all parasites and 1.37 (95% CI: 0.95, 1.79) for gametocytes. This excess co-infection risk was partially attributable to malaria infections clustering in some villages. *Pf-Pv-Pm* triple infections were four times more frequent than expected by chance alone, which could not be fully explained by infections clustering in highly exposed individuals. The effect of co-infection on parasite density was

**Funding:** This work was carried out with funding awarded to IF from the Swiss National Science Foundation (grants no. 310030-134889 and 310030-159580), NIH NIAID International Centres of Excellence in Malaria Research South West Pacific (U19 AI089686) and Asia Pacific (U19 AI129392-01) awarded to IM. Field work was supported by the TransEPI consortium funded by the Bill & Melinda Gates Foundation awarded to IM. IM (GNT11155075) and LJR (GNT1161627) are supported by NHMRC Fellowships. MTW is supported by the French Government's Laboratoire d'Excellence "Integrative Biology of Emerging Infectious Diseases" (Investissement d'Avenir grant n°ANR-10-LABX-62-IBEID), and the INCEPTION program (Investissement d'Avenir grant ANR-16-CONV-0005). The funders had no role in study design, data collection and analysis, decision to publish, or preparation of the manuscript.

**Competing interests:** The authors have declared that no competing interests exist.

analyzed by systematic comparison of all pairwise interactions. This revealed a significant 6.57-fold increase of *Pm* density when co-infected with *Pf*. *Pm* gametocytemia also increased with *Pf* co-infection.

## Conclusions

Heterogeneity in exposure to mosquitoes is a key epidemiological driver of *Plasmodium* co-infection. Among the four co-circulating parasites, *Pm* benefitted most from co-infection with other species. Beyond this, no general prevailing pattern of suppression or facilitation was identified in pairwise analysis of gametocytemia and parasitemia of the four species.

## Trial registration

This trial is registered with ClinicalTrials.gov, Trial ID: NCT02143934.

## Author summary

The majority of malaria research focuses on the *Plasmodium falciparum* and *P. vivax* parasite species, due to their large public health burden. The epidemiology of *P. malariae* and *P. ovale* parasites has been comparatively neglected, due to a lack of research tools, most notably diagnostics. We present new molecular diagnostic assays for detecting *P. malariae* and *P. ovale* gametocytes, the sexual stage of the malaria parasite transmitted to mosquitoes. These assays were applied to samples collected in Papua New Guinea, a rare region with high transmission of the four major malaria parasite species. Patterns of co-infections were characterized accounting for interactions between pairs and triples of parasites. Heterogeneity in exposure to mosquito bites was identified as a key driver of patterns of co-infection. The effect of co-infection on parasite density was analyzed by systematic comparison of all pairwise interactions. The most significant within-host interaction of parasites was the large increase in *P. malariae* parasite density due to co-infection with *P. falciparum*. This finding was replicated for *P. malariae* gametocytes (but did not attain statistical significance due to low sample numbers) suggesting that co-infection provides a key transmission advantage to *P. malariae*.

## Introduction

Interactions of multiple *Plasmodium* species co-infecting a single human host have been investigated since the early days of the malariological literature. Data from malariatherapy in syphilis patients suggested interaction between different *Plasmodium* species [1,2], for instance, one species dominated the other after simultaneous inoculations with *P. falciparum* (abbreviated to *Pf*) and *P. vivax* (abbreviated to *Pv*) or with *Pv* and *P. malariae* (abbreviated to *Pm*). Since then, there have been many reports on *Plasmodium* species interactions, mainly targeting *Pf-Pv* and *Pf-Pm* co-infections [3–6].

In many previous reports (summarized in **Table A in S1 Text**), one species dominated the other. However, the outcome of within-host co-infection may not only consist of within-host competition and suppression of one of the infecting species. One of the species may also be favored by facilitation, or both species may be positively or negatively affected [6,7]. Although many earlier reports indicated interactions between different *Plasmodium* species, it was not

obvious if these were positive or negative. Interactions may be experimentally measured as changes in parasite density caused by the presence of a co-infecting genotype. A positive association between two parasite species indicates that co-infection is more likely than expected under the hypothesis of independence, and vice versa for negative associations. Heterogeneity in exposure to infectious mosquitoes poses a further challenge to understanding between-species interactions. For instance, two species may be frequently co-observed because they are transmitted by the same mosquito, rather than because of facilitating interactions.

As *Pm* and *Pf* are sympatrically distributed in many endemic areas, mixed infections of both species are frequent [8–11]. Although *P. ovale* sp. (abbreviated to *Po*) is often detected in mixed infections with the other three major *Plasmodium* species in the Pacific, Asia and Africa [8,10,11], the low prevalence of this species limits meaningful statistical analyses of interactions. Several studies describe positive associations of *Pf* with *Pm* and/or *Po* [3,4]; while in contrast others describe seasonal fluctuations of *Pf* prevalence and *Pm* density which can be interpreted as negative interaction, i.e. suppression of *Pm* by *Pf* [3]. Many of the published prevalence rates for *Pm* were determined by light microscopy (LM) and therefore other low density co-infecting species may be overlooked. This compromises any comparison with more recent PCR-based data, particularly when assessing the effects of co-infection on prevalence and density of the other species.

Suppression occurring in *Pf-Pv* co-infection was suggested by a systematic review of LM data [6]. Considerable previous research, particularly on *Pf-Pv* co-infection, has focused on the fact that species co-infection may also involve immunological interaction [12]. The host's allocation of immune response against co-infecting parasites might involve complex trade-offs, for example, between investment in defense against one parasite species on the cost of controlling the other species. Complex interactions likely depend on life-history traits of both co-infecting species, such as different host cell requirements or length of life cycle or duration of a natural infection. Several studies reported that *Pv* and *Pm* infections reduce the severity of falciparum malaria in co-infections [13–15]. These findings gave rise to speculation that *Pv* infection, or in general infection with a less virulent parasite [13], may protect against subsequent severe *Pf* malaria by inducing cross-species immunity and acting as a hypothetical natural vaccine in areas where both parasites co-exist [13]. However, other studies found that *Pf-Pv* co-infections were present in severe malaria cases, arguing against *Pv* protecting against *Pf* infection [16,17].

Gametocytes, the sexual stages of *Plasmodium* species, are the only life stages transmissible to mosquito vectors and therefore essential for onward transmission. There is limited knowledge on the effects of multiple species co-infection on gametocyte production and transmission [18] (summarized in **Table B in S1 Text**).

Many previous studies are limited by the low sensitivity of LM. When no molecular detection is used, low-density *Plasmodium* species co-infection may fall below the limit of detection of LM and remain undetected. Here we present molecular diagnostic data on interactions among the four major human *Plasmodium* species. In addition to quantifying the parasitemia of all species by qPCR, the carriage of sexual stages (gametocytes) of each species was measured by RT-qPCR to assess effects of co-infection on onward transmission. To detect gametocytes of all four species, we developed molecular assays for *Pm* and *Po* gametocyte detection, for which no assays are published to date. These assays were based on the *Pm* and *Po* orthologues of *pfs25* and *pvs25* genes and used in combination with earlier developed *Pf*- and *Pv*-specific RT-qPCRs [19].

The present study aims to identify the effect of co-infection on parasite density and gametocyte carriage for all possible combinations of the four major human *Plasmodium* species using state of the art molecular diagnostics [19].

## Methods

### Ethics statement

Clinical trial registration: ClinicalTrials.gov NCT02143934. Human subjects: The study received ethical clearance from the Papua New Guinea Institute of Medical Research Institutional Review Board (0908), the PNG Medical Advisory Committee (09.11), the Ethics Committee of Basel 237/11 and was conducted in full concordance with the Declaration of Helsinki. Written informed consent was obtained from the parents/guardians of all children enrolled in the study.

### Study site and population

A comprehensive investigation of co-infection was possible through a unique setting of four sympatric *Plasmodium* species in Papua New Guinea (PNG). At the study site in East Sepik Province, *Pv* prevalence is among the highest reported globally [20,21]. *Pf* in PNG can locally reach holo-endemic transmission levels otherwise only seen in sub-Saharan Africa [22]. *Pm* and *Po* are often found as mixed-species infections in PNG [18]. During the study period, East Sepik Province had amongst the highest rates of malaria prevalence in PNG [23,24].

Nucleic acid samples were collected from the Albinama cohort study conducted between 2009 and 2010 in villages of Maprik District, East Sepik Province, PNG. Full details of the study are provided in Robinson *et al* [21] and a CONSORT checklist is provided in **Fig A in S1 Text**. In brief, 504 children aged 5–10 years were recruited and randomized to receive antimalaria treatment targeting blood-stage parasites, or treatment targeting both blood- and liver-stages. Children were actively monitored for infection by qPCR and illness for a total of 32 weeks. Finger-prick blood samples were taken every two weeks for the first 12 weeks and every 4 weeks from week 14–32.

### Detection of blood-stage parasites and gametocytes

Previously published data was incorporated in the present analysis, namely qPCR-based and RT-qPCR-based detection of *Pf*, *Pv*, *Pm* and *Po* parasitemia, as well as RT-qPCR-based *Pf* and *Pv* gametocytemia [25]. Data on *Pm* and *Po* gametocytemia was newly generated by RT-qPCR using previously extracted RNA samples stored at -80C. *Plasmodium* species blood-stage infections were detected using a generic qPCR assay to detect *Plasmodium* infection by any species [19,26], followed by species-specific (*Pf*, *Pv*, *Pm* and *Po*) qPCRs targeting the *18S rRNA* genes [27]. The *Po* qPCR detects both *Po curtisi* and *Po wallikeri*. *Pf* and *Pv* gametocytes were detected and quantified using RT-qPCRs targeting the transcripts of *pfs25 or* orthologues [19]. qPCR assays detect DNA of all parasite stages present in the blood, including genomic DNA of gametocytes. Because asexual parasite stages contribute the overwhelming fraction of blood stages (except for rare cases of persisting gametocytes but cleared asexuals), we used the term "asexual" to denote the total blood stage parasites based on qPCR. This asexual fraction we contrast with the pure gametocyte fraction obtained by detecting stage-specific gametocyte RNA.

For detection and quantification of *Pm* and *Po* gametocytes, two RT-qPCR assays targeting *pms25* and *pos25* (PlasmoDB: PmUH01_10042200 and PocGH01_06024100, respectively) transcripts were developed and validated. *Pfs25* and *pvs25* are among the highest transcribed genes in mature gametocytes [26]. Primers as well as HEX-BHQ1-labelled and FAM-MG-B-EQ-labelled probes of *pms25* and *pos25*, respectively, were selected within regions of maximal diversity to its orthologues. Owing to the lack of *pos25* and *pms25* sequences in GenBank, the target region was amplified and sequenced from field isolates to identify conserved regions. Sequences of oligonucleotides as well as reaction mixes and thermal profiles are shown in

**Tables C-E in S1 Text**. Although there is evidence that *P. ovale wallikeri* and *P. ovale curtisi* are distinct species that are sympatric and do not recombine [28], in this study we analyze these parasites as a single species complex *P. ovale* sp.

Analytical specificity of the *pms25* and *pos25* RT-qPCRs was assessed both in silico using sequences of human *Plasmodium* species identified by PlasmoDB and BLAST searches. Specificity was evaluated experimentally using *Pf*, *Pv*, and *Po* or *Pm* gDNA and human gDNA from a malaria-free anonymous blood donor. The new gametocyte-specific *pms25* and *pos25* RT-qPCR assays did not show any cross-reactivity and were negative for human and non-*malariae* or non-*ovale Plasmodium* DNA samples. All parasite DNAs from field samples that were selected for this validation were negative for *Pm* or *Po* as confirmed by 18S qPCR.

Analytical sensitivity and RT-qPCR efficiency were validated on dilution rows of synthesized pCR 2.1-TOPO TA vectors (Invitrogen, Switzerland) containing inserts of the respective *pms25* or *pos25* amplicons. Details on plasmid dilution rows and performance of RT-qPCR assays are presented in **Table F in S1 Text**. Sensitivity was defined by the lowest concentration of the standard plasmid whereby at least 50% of the replicates were positive (see **Tables G-H in S1 Text**). Amplification efficiency was calculated for each assay as Efficiency = $10^{(-1/Slope)}$ -1. Details on intra- and inter-assay coefficient of variation (CV) of the *pms25* and *pos25* RT-qPCR assays are shown in **Tables J-K in S1 Text**.

## Statistical analysis

Let $X_i$ be the asexual or gametocyte parasite prevalence of $i \in \{Pf, Pv, Pm, Po\}$. Denote $X_{ij}$ as the observed co-infection prevalence of parasites $i$ and $j$. If parasites $i$ and $j$ are independently distributed, the expected co-infection prevalence is $X_i X_j$. The excess co-infection risk of parasites $i$ and $j$ is defined as $\frac{X_{ij}}{X_i X_j}$. The excess co-infection risk is analysed across all pairs of parasites using the following linear regression:

$$x_{ij} \sim x_i x_j \ i \neq j \in \{Pf, Pv, Pm, Po\}$$

The intercept is fixed at 0, and the estimated slope β is the average excess co-infection risk across all pairs of parasites.

Let $X_{ijk}$ be the observed triple infection prevalence of parasites $i$, $j$ and $k$. The excess risk of triple infection can be defined as $\frac{X_{ijk}}{X_i X_j X_k}$. The excess triple infection risk can be analysed using linear regression as follows:

$$x_{ijk} \sim x_i x_j x_k \ i \neq j \neq k \in \{Pf, Pv, Pm, Po\}$$

The intercept is fixed at 0, and the estimated slope γ is the average excess triple infection risk across all triples of parasites. If excess triple infection risk was entirely attributable to the pairwise clustering of parasites, we would expect $\hat{\gamma} = \hat{\beta}^2$. If triple infections are more common than can be explained by pairwise co-infections, then we expect $\hat{\gamma} > \hat{\beta}^2$.

Denote $Y_i$ as the density of asexual parasites or gametocytes where $i \in \{Pf, Pv, Pm, Po\}$. The interaction between the densities of parasites $i$ and $j$ is analysed using the following linear regression:

$$log(Y_i) \sim \delta_{ij} + \theta$$

where $\delta_{ij}$ = 1 in the case of co-infection, and $\delta_{ij}$ = 0 otherwise. θ is a covariate accounting for either pre-treatment status; post-treatment with blood-stage drugs; or post-treatment with blood-stage drugs plus PQ. Data were analysed using R v3.5.3 statistical software.

## Results

### Co-infections of 4 *Plasmodium* species in PNG

Co-infection patterns for *Pf*, *Pv*, *Pm* and *Po* asexual parasites and gametocytes are shown in **Fig 1** for the 504 pre-treatment samples. The pattern of co-infections was also analyzed for all 5561 samples collected during the entire cohort study (**Fig B in S1 Text**). For the pre-treatment samples collected at enrolment there was substantial heterogeneity in asexual parasite and gametocyte prevalence between the five neighboring villages of the cohort study[20]. Compared to the reference village of Albinama, the village of Bolumita had significantly higher prevalence of asexual parasites and gametocytes for both *Pf* and *Pm* (**Table L in S1 Text**). With 504 samples collected at enrolment, no other associations were statistically significant (age, sex, bed net use). When the analysis was extended to 5561 observations from the entire cohort study, a number of associations observed at enrolment became statistically significant (**Table O in S1 Text**).

### Estimating excess co-infection risk

In the pre-treatment samples, *Pf* prevalence was 23.8% (120/504) and *Pv* prevalence was 47.4% (239/504). If these parasites were randomly distributed, we would expect co-infection

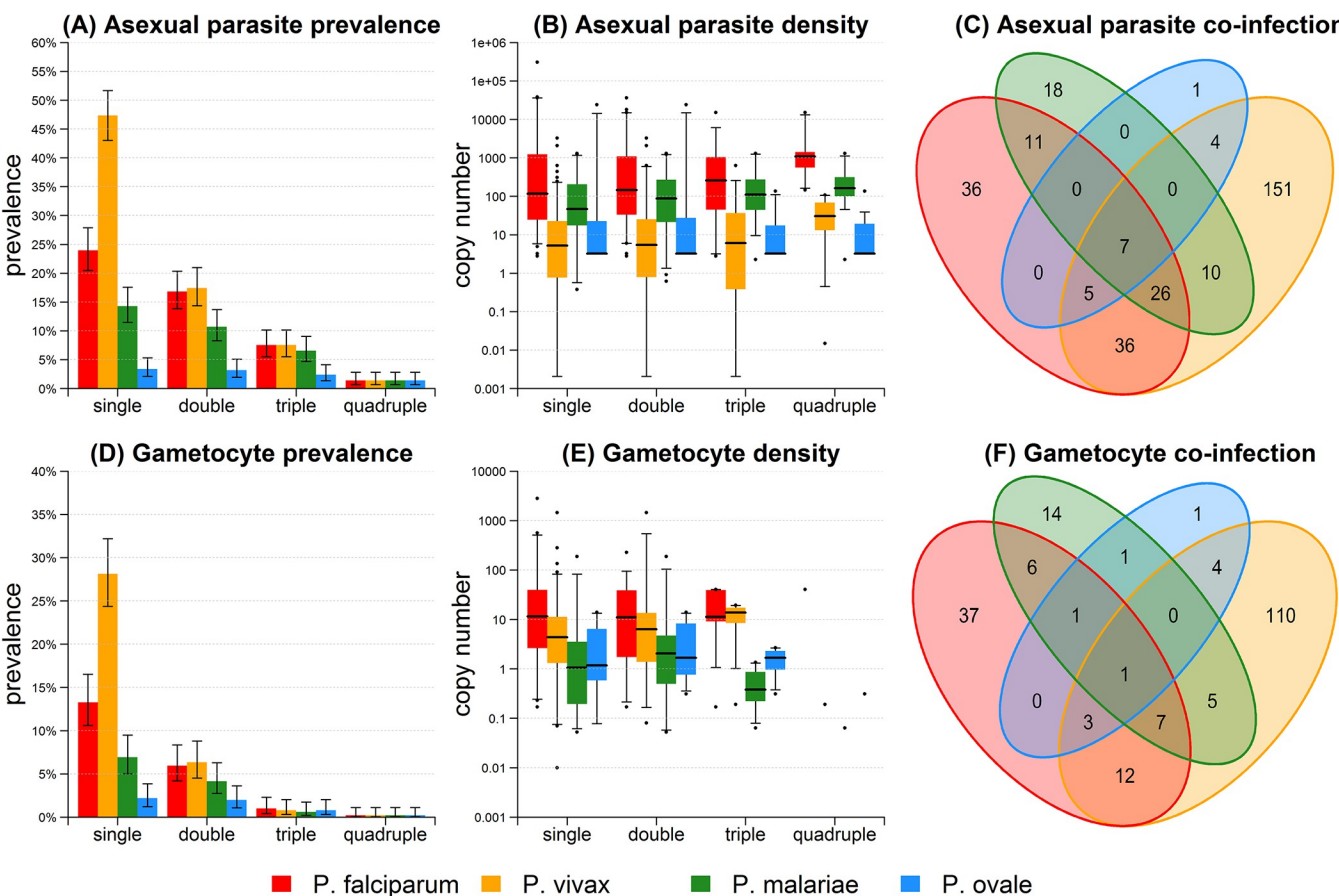

**Fig 1. Malaria co-infection in pre-treatment samples (*n* = 504).** (A) Co-infection prevalence of asexual parasites. Double infection with *Pf* denotes the proportion of samples PCR positive for *Pf* and at least one other species. Triple infection with *Pf* denotes the proportion of samples PCR positive for *Pf* and at least two other species. Other bars are similarly defined. 95% confidence intervals were calculated using Wilson's binomial method. (B) Asexual parasite density in co-infected samples. (C) Venn diagram of asexual parasite co-infection. (D) Co-infection prevalence of gametocytes. (E) Gametocyte density in co-infected samples. (F) Venn diagram of gametocyte co-infection.

prevalence of 0.238 * 0.474 = 11.3%. However, in the data we observe co-infection prevalence of 14.7% (74/504). The excess risk of *Pf* and *Pv* co-infection is therefore 14.7%/11.3% = 1.29. When we account for all six pairwise combinations of co-infection as shown in **Fig 2**, we estimate the average risk of excess co-infection as 1.38 (95% CI: 1.09, 1.68).

The analysis in **Fig 2** was applied to the combined data from all five villages. Excess co-infection risk could be attributable to unaccounted heterogeneity, for example if malaria infections cluster in some villages. The analysis of asexual parasite co-infection was repeated with stratification by village in **Fig 3A**, with an estimate of excess co-infection risk of 1.11 (95% CI; 1.07, 1.16). Therefore, a proportion of the excess co-infection risk is due to the concentration of malaria of all species within high transmission villages. A similar pattern was observed for gametocyte co-infection (**Fig 3C**).

This approach can be extended to account for triple infection. Considering *Pf* (120/504 = 23.8%), *Pv* (239/504 = 47.4%) and *Pm* (72/504 = 14.3%) in the pre-treatment samples, if parasite infection was independent, we would expect 0.238*0.474*0.143 = 1.6% of individuals with all three parasites. Instead, we observe 33/504 = 6.5% with *Pf*, *Pv* and *Pm* asexual parasites. This is 6.5/1.6 ~ 4 times higher than expected by chance. Using a statistical model to account for the four possible combinations we estimated that triple infections are 2.42 (95% CI; -1.62, 6.47) times more frequent than expected by chance alone (**Table 1**). This association is not significant, partially due to being based on four observations. We can further apply the analysis to 20 observations of triple infection by stratifying by the five villages (**Fig 3B**) in which case we estimate a significant excess risk of triple infections of 1.51 (95% CI; 1.38, 1.64).

A hypothesis for the excess prevalence of triple infection is that it is a consequence of the excess risk of co-infection with pairs of parasites. For the village-stratified data, the excess risk of co-infection with two parasites was 1.11. If the observed patterns were due to heterogeneity measured in this way, we would expect an excess risk of triple infection of 1.11*1.11 = 1.23. Thus, we can estimate an excess risk of triple infection adjusted for heterogeneity of 1.51/1.23 = 1.22 (95% CI; 1.11, 1.32) (**Table 1**). This implies that triple infections are even more common than can be explained by clustering of infections in highly exposed individuals.

## Effect of co-infection on parasite density

For the eight types of parasites measured (asexual parasites and gametocytes to *Pf*, *Pv*, *Po*, and *Pm*) there are 56 possible pairwise interactions. **Fig 4** presents a systematic comparison of all pairwise interactions for the pre-treatment samples. For example, *Pm* asexual parasites had a geometric mean density of 14 copy numbers/µL in the absence of *Pf* parasites. However, when co-infected with *Pf* asexual parasites the geometric mean of *Pm* asexual parasite density increased to 94 copy numbers/µL. Therefore, *Pf* co-infection was associated with a 6.57 (95% CI; 2.93, 14.75; P = 2.1 x 10$^{-5}$) increase in *Pm* density. This association was still significant after adjusting for multiple hypothesis testing (P = 0.0005). One other association remained significant after correction for multiple hypothesis testing: the density of *Pv* asexual parasite infection was greater when *Pv* gametocytes were also detected. This is likely a consequence of the high degree of correlation between *Pv* asexual and gametocyte densities, and the challenge of detecting low density infections [25]. Although not significant after multiple hypothesis testing, we observed that *Pv* and *Pm* asexual densities were higher in co-infection with *Po* compared to *Pv* densities in the absence of *Po*.

Following the finding on *Pm* densities being determined by co-infections, an analysis of the effects of *Pm* co-infection over time was performed. Enrolment prevalence of *Pm* asexual infection was 14.3% (**Fig 5A**) and 6.9% for gametocytes (**Fig 5B**). After treatment, prevalence dropped to zero and thereafter rose very slowly over time. A very high proportion of *Pm*

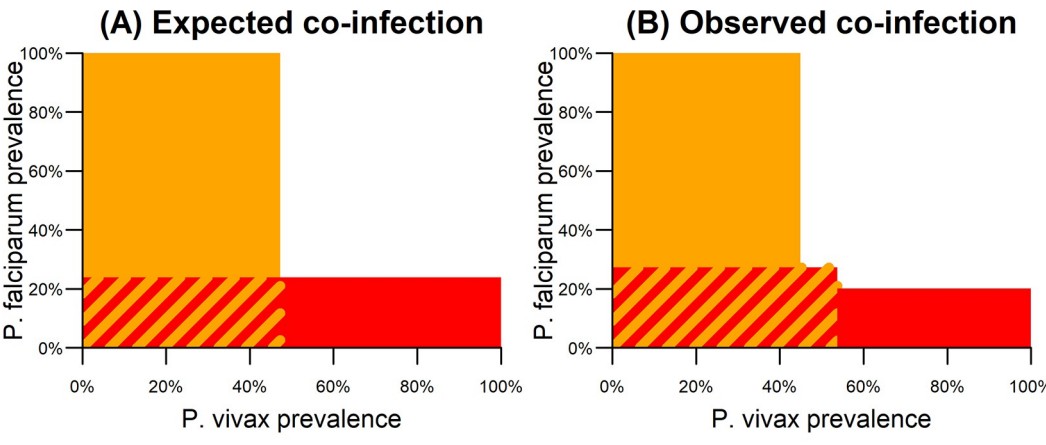

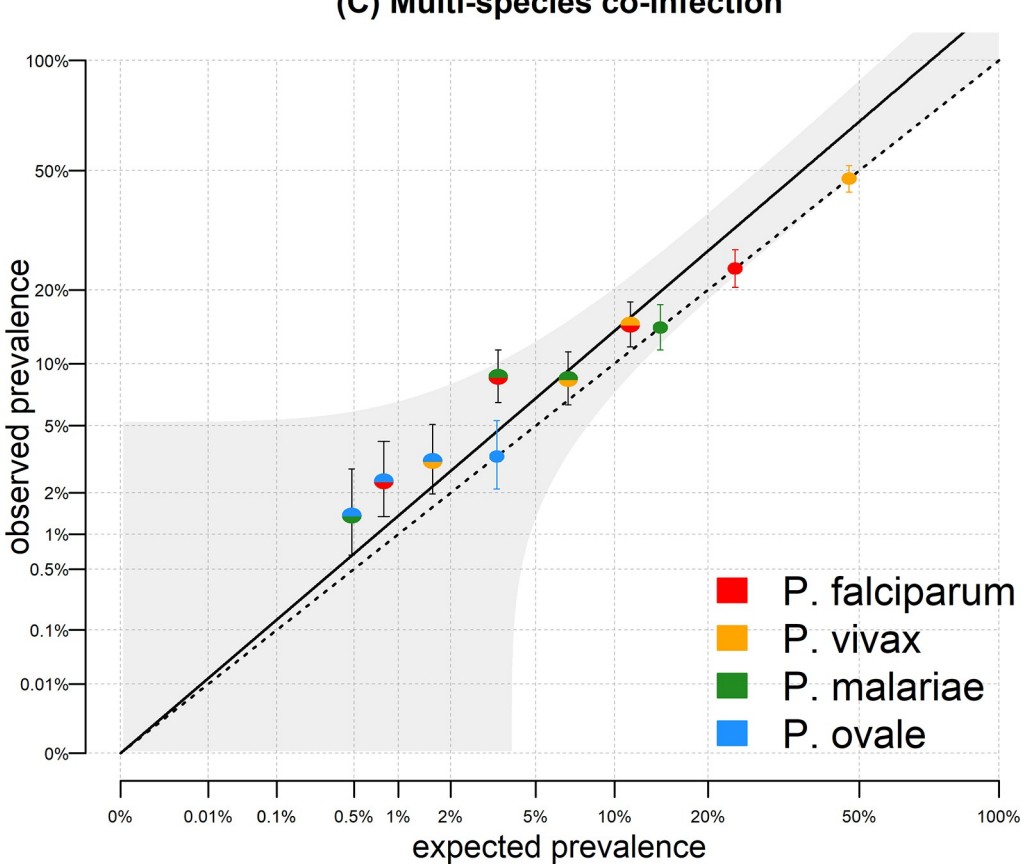

**Fig 2. Excess asexual parasite co-infection in pre-treatment samples.** **(A)** In the pre-treatment samples, Pf asexual prevalence was 23.8% (120/504) and Pv asexual prevalence was 47.4% (239/504). If these parasites were randomly distributed, we would expect co-infection prevalence of 0.238 * 0.474 = 11.3% (red and yellow striped region). **(B)** Co-infection prevalence of 14.7% (74/504) was observed, in excess of what is expected by random mixing. **(C)** Expected versus observed co-infection prevalence for the six pairwise combinations of asexual parasites. The dashed line represents the scenario where observed co-infection prevalence equals expected prevalence. Mono-coloured points denote observed prevalence rates of the four species. The multi-coloured data points fall above this line. The solid line denotes a regression model fitted through these points, with 95% confidence intervals shown in grey.

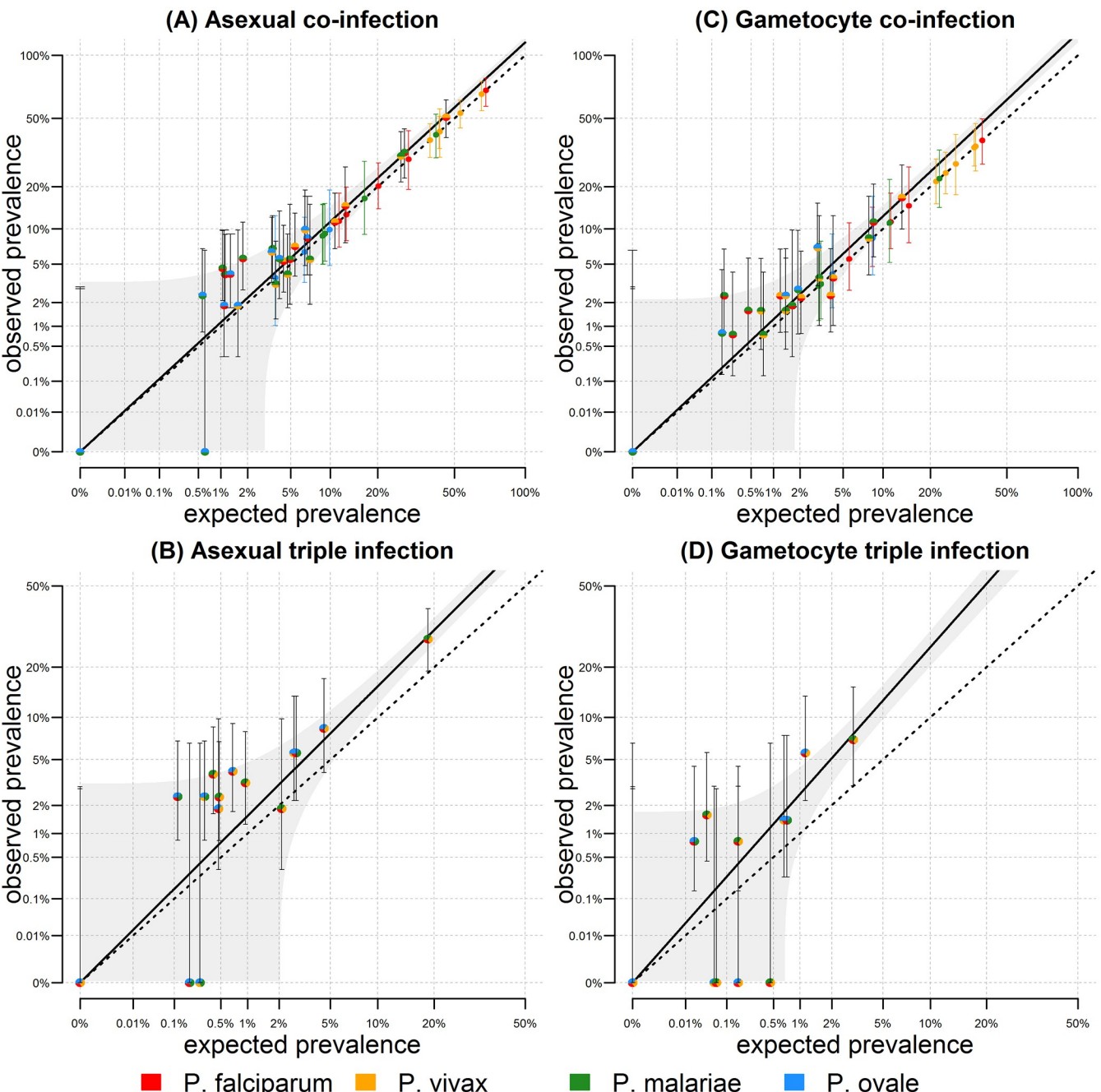

**Fig 3. Co-infection in pre-treatment samples stratified by village. (A)** For the six pairwise combinations of two malaria species, the observed and expected co-infection asexual prevalence is plotted for each of the five villages. The dashed line represents the scenario where observed co-infection prevalence equals expected prevalence. The multi-coloured data points tend to fall above this line. The solid line denotes a regression model fitted through these points, with 95% confidence intervals shown in grey. **(B)** Observed and expected triple infection. If the prevalence of $Pm$ is $X_{Pm}$, then the expected prevalence of $Pf$, $Pv$ and $Pm$ co-infection is $X_{Pf} * X_{Pv} * X_{Pm}$. The multi-coloured points fall above the dashed line indicating greater observed than expected prevalence. The solid line denotes a regression model fitted through these points. **(C)** Gametocyte co-infection. **(D)** Gametocyte triple infection.

infections also occurred as co-infection with another *Plasmodium* species. Before treatment, 61.1% (49.6%, 71.5%) of *P. malariae* infections were observed to be co-infected with *P. falciparum* (**Fig 5C**). Notably, this is greater than the expected proportion of *P. malariae* infections

**Table 1. Estimated risk of co-infection in pre-treatment samples.** *The excess risk of triple infection accounting for the known pairwise clustering due to heterogeneity (e.g. $1.26 = 2.42/1.38^2$).

| | Asexual parasites | | Gametocytes | |
|---|---|---|---|---|
| | risk | P value | risk | P value |
| *Pre-treatment samples* | | | | |
| excess risk of 2nd infection | 1.38 (1.09, 1.68) | 0.0489 | 1.38 (0.96, 1.80) | 0.13 |
| excess risk of 3rd infection | 2.42 (-1.62, 6.47) | 0.54 | 4.67 (-0.77, 10.12) | 0.28 |
| heterogeneity adjusted excess risk of 3rd infection * | 1.26 (-0.85, 3.37) | 0.82 | 2.44 (-0.40, 5.29) | 0.39 |
| *Pre-treatment samples; stratified by village* | | | | |
| excess risk of 2nd infection | 1.11 (1.07, 1.16) | $1.4 \times 10^{-5}$ | 1.22 (1.13, 1.32) | $5.5 \times 10^{-5}$ |
| excess risk of 3rd infection | 1.51 (1.38, 1.64) | $7.3 \times 10^{-8}$ | 2.42 (2.05, 2.79) | $1.2 \times 10^{-7}$ |
| heterogeneity adjusted excess risk of 3rd infection * | 1.22 (1.11, 1.32) | 0.00046 | 1.61 (1.37, 1.86) | $6.3 \times 10^{-5}$ |

co-infected with *P. falciparum* under an assumption of random mixing. After treatment, the co-infection proportion increases over time, albeit with substantial variation.

## Discussion

To investigate the prevalence of gametocytes of the four major *Plasmodium* species in humans and their within-host interactions, we developed new assays for *Pm* and *Po* gametocyte quantification to complement existing methods for *Pf* and *Pv* gametocytes. To our knowledge, this is the first study that analyzes quantitative molecular data of both asexual parasite stages and gametocytes for these four parasite species.

The present study showed an average risk of excess co-infection of 1.38 when accounting for all six pairwise combinations of co-infection in the pre-treatment samples from all five study villages. This excess in co-infections is partially due to transmission heterogeneity, i.e. when malaria infections cluster in some of the villages. After stratifying by village, an estimated excess risk of co-infection of 1.11 was found. Thus, a proportion of the excess co-infection risk was attributable to the concentration of all *Plasmodium* species within high transmission villages. A similar pattern was observed for gametocyte co-infection.

Several factors may contribute to a heterogeneous parasite distribution, such as host susceptibility, mosquito vector ecology and transmission seasonality [29]. Previous studies in PNG reported significant heterogeneity in malaria transmission that was attributed to the geographic diversity within the country and local population structure [22,23,30]. Earlier analyses of this study have found considerable micro-spatial heterogeneity in malaria transmission among both neighboring communities and individual children from the same village [20].

In mixed-species co-infections the number of transmission stages produced may be up- or downregulated by within-host interactions. For instance, any signal from a co-infecting or dominant species could induce gametocyte production. Such a signal was shown to facilitate induction of gametocytogenesis: lysophosphatidylcholine (LysoPC) is taken up by *Pf* and triggers gametocyte production depending on LysoPC plasma levels [31]. Since all four human *Plasmodium* species harbor the effector gene *gametocyte development protein 1* (*gdv1*), LysoPC-driven facilitation of a co-infecting species' investment in gametocytes may be expected. On the contrary, a so far unidentified signal could down-regulate the gametocyte production of a co-infecting species and thus influence gametocyte carriage.

A striking outcome of this study was that *Pm* density increased with any additional co-infection. Co-infections with *Pf* asexual parasites lead to a significant 6.57-fold increase in *Pm* asexual parasite density. It was also observed that in co-infection with *Pm*, the mean asexual and gametocyte densities of any other co-infecting species were reduced compared to those in

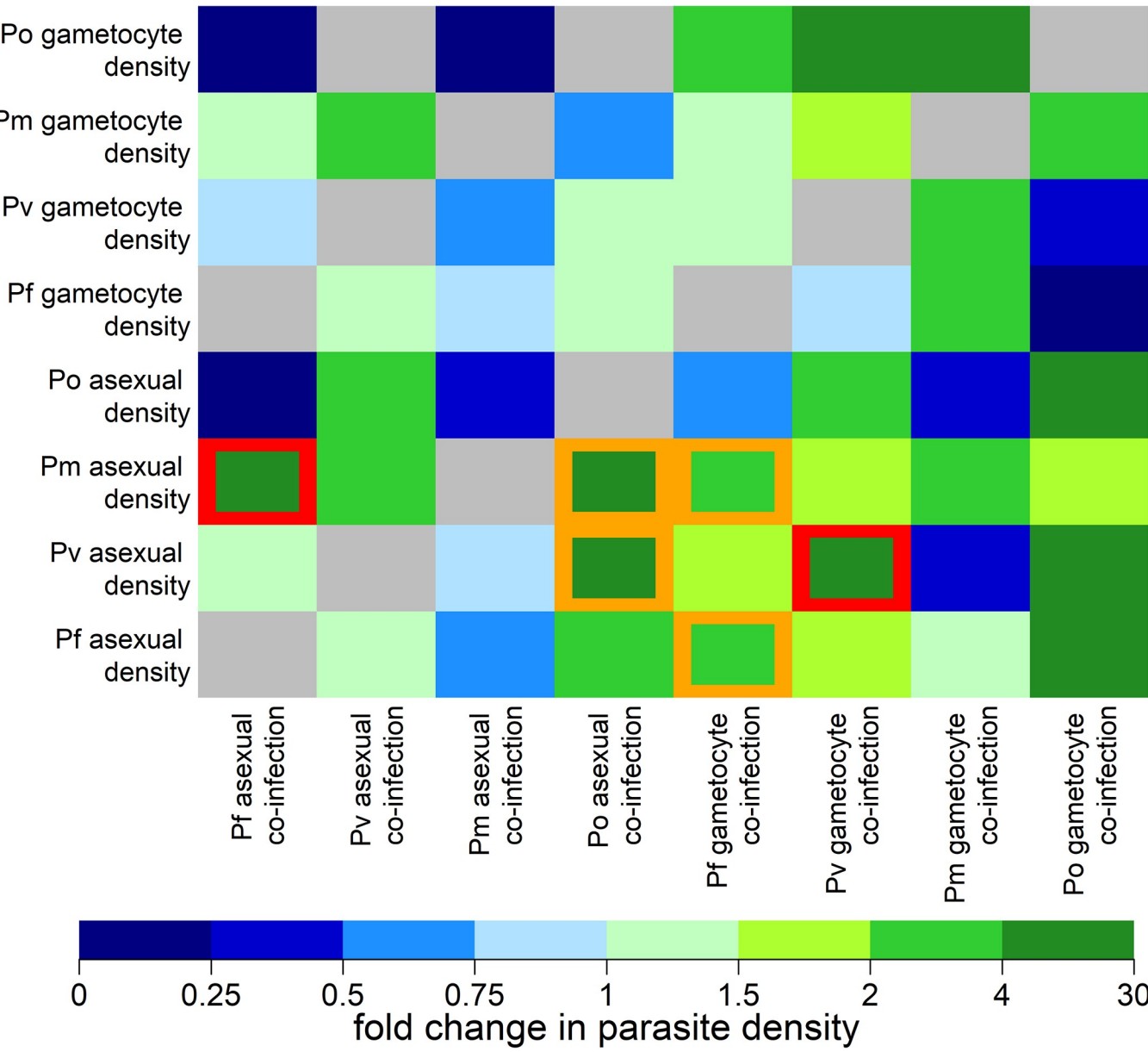

**Fig 4. Effect of co-infection on parasite density in pre-treatment samples.** The y-axis denotes the measured parasite density and the x-axis denotes the confounding effect of co-infection. Each square denotes the fold change in parasite density due to co-infection. For example, for *Pm* asexual parasites, co-infection with *Pf* asexual parasites leads to a 6.57 (2.9, 14.8) fold increase in *Pm* asexual parasite density. Grey squares denote interactions where it was not possible to estimate an effect. Otherwise, all estimated effects are presented regardless of statistical significance. Orange squares denote significant associations with P values < 0.05. Red squares denote significant associations with P values < 0.05 after the Benjamini-Hochberg adjustment for multiple hypothesis testing.

single-species infections, although these effects were not statistically significant. These data could suggest that *Pm* thrives on the cost of co-infections and outcompetes these. In line with these observations in PNG, findings from retrospective analyses of malaria therapy data showed that primary *Pm* infections were protective against secondary *Pf* infections [32]. To investigate whether *Pm*-specific infection dynamics could be a driver of the observed effect, a longitudinal analysis of *Pm* infections (**Fig 5**) was performed. *Pm* prevalence increased very slowly over time—consistent with a low force of infection combined with a long duration of

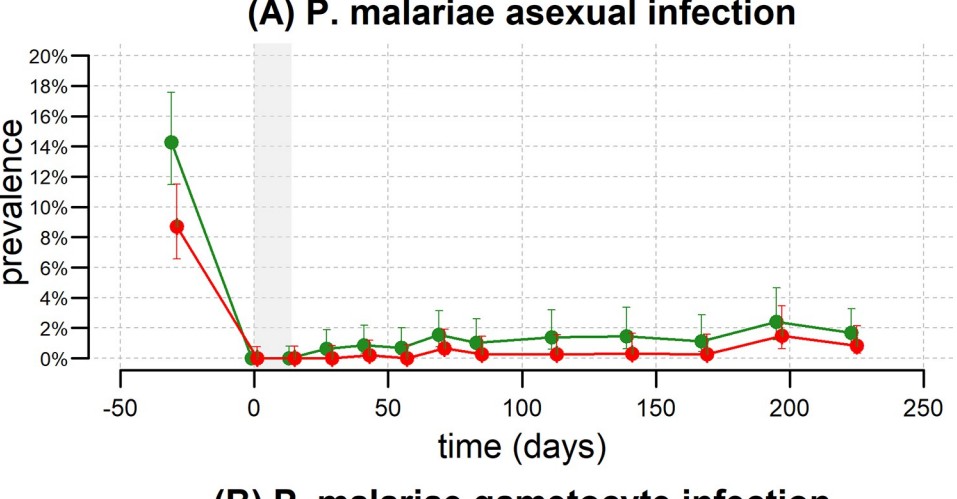

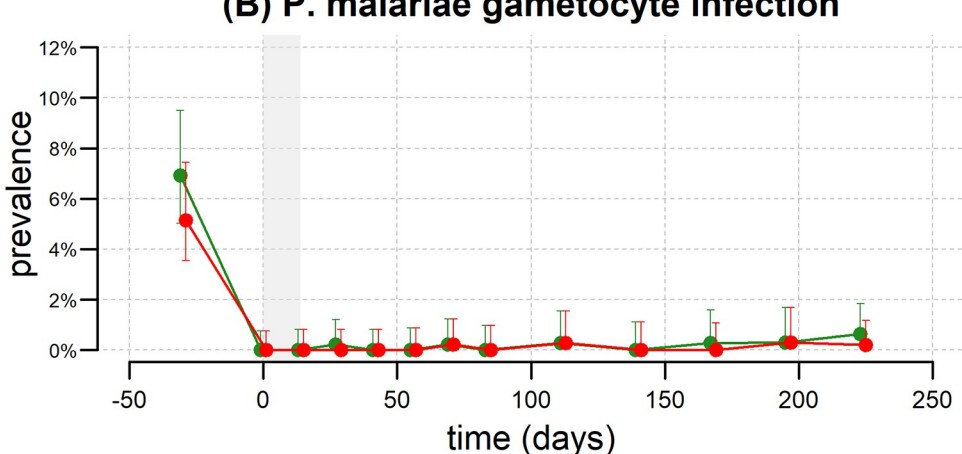

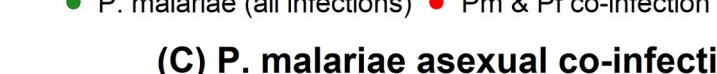

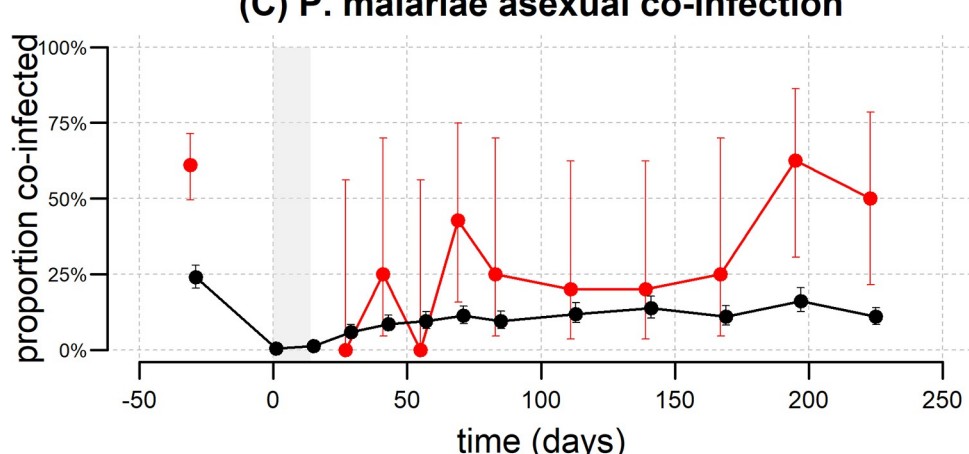

**Fig 5. Longitudinal analysis of *P. malariae* and the effect of co-infection.** The prevalence of (**A**) *P. malariae* asexual parasites, and (**B**) *P. malariae* gametocytes. The prevalence of all *P. malariae* infections is shown in green, and the prevalence of *P. malariae* and *P. falciparum* co-infection is shown in red. The grey shaded region denotes the period of treatment. 5561 samples were included over the entire time period. 123 samples were positive for *P. malariae* asexual parasites, and 61 of these were co-infected with *P. falciparum* asexual parasites. 43 samples were positive for *P.*

*malariae* gametocytes, and 18 of these were co-infected with *P. falciparum* gametocytes. **(C)** The observed proportion of *P. malariae* asexual infections that are co-infected with *P. falciparum* is shown in red. The expected proportion of *P. malariae* infections co-infected with *P. falciparum* under an assumption of random mixing is shown in black.

infection. This is consistent with *Pm* causing chronic infections that can last for years[8], possibly facilitated by interactions with other parasites.

The *Pm* and *Po* gametocyte detection assays were designed based on the assumption of functional similarity because of sequence similarity to the highly expressed *pfs25* orthologue i.e. *pms25* and *pos25* show similar high expression to *pfs25*. Because no transcriptome data for *pms25* and *pos25* are yet available to assess accurate transcription levels of these orthologues, it cannot be excluded that more suitable markers with even higher expression levels could exist. Our validation of the *pfs25* orthologues showed good performance and sensitive detection of the *pms25* and *pos25* transcripts with our assays.

A major technical limitation of the study was that blood for RNA extraction was collected on filter paper and later stored in TRIzol for several years. RNA sample storage in TRIzol was shown to be suboptimal compared to direct transfer of blood into a RNA-stabilizing reagent [19]. Therefore, gametocyte densities could have been underestimated or remained undetected due to suboptimal sampling and storage, leading to RNA degradation. Another limitation consisted in the narrow age range of study participants (i.e. children aged 5–10 years). Thus the data may not be representative of the total population. We did not analyze morbidity because there were not enough clinical cases of malaria to assess inter-species interactions with sufficient statistical power. A further challenge of this study was the limited number of samples positive for *P. malariae* or *P. ovale* sp.. These limited sample numbers were also a barrier to the analysis of the sub-species of *P. o. curtisi* and *P. o. wallikeri*.

Our analysis of species interaction was essentially cross-sectional focusing on the pre-treatment samples of the cohort study. A full longitudinal analysis would be more powerful, but would need to account for the PQ treatment given randomly to 50% of participants after the baseline sample. We make the simplifying assumption that PQ should not affect co-infection analysis of the follow-up samples. However, in the PQ group newly appearing infections are predominantly derived from mosquito infections, while in the placebo group, newly appearing *Pv* or *Po* parasites could be either relapses or true new infections.

A further limitation is that the analyses presented did not account for individual-level heterogeneity in exposure, which would have been possible in *Pf* or *Pv* infected individuals by adjusting for molecular force of blood-stage infection ($_{mol}$FOB) [20]. This could potentially reduce some of the remaining heterogeneity, which was not removed by the village-level adjustment of our analysis.

To our knowledge, this is the first study that analyses the effect of co-infection on both parasite density and gametocyte carriage for all possible combinations of the four major human *Plasmodium* species using molecular data of asexual parasite stages and gametocytes. We found an excess risk of co-infection that was higher than expected by chance alone. Similarly, triple infections were more common than expected from clustering of infections in highly exposed individuals. We attributed these observations partly to heterogeneity in exposure. The effects of co-infection among *Plasmodium* species did not follow a generalized pattern. Instead, species-specific effects of within-host interaction seem to act in opposite directions. The clearest picture emerged from the total parasite density data, but less so from gametocyte data. *Pm* benefitted most from co-infections with any other *Plasmodium* species, as its asexual and gametocyte densities were increased in co-infections.

Parasite epidemiology and biology is complex, leading to both positive and negative effects on parasite prevalence and density for all pairs of co-infecting species. Despite their limitations

for definitively inferring facilitation or suppression, the molecular assays and statistical analyses presented here allow for improved understanding of malaria parasite co-infection.

## Supporting information

**S1 Text. Additional file on molecular methods, assay conditions for *P. malariae* and *P. ovale* sp. gametocyte-specific RT-qPCR targeting pms25/pos25, and additional results from sensitivity analyses.**
(DOCX)

## Acknowledgments

We sincerely thank the children, their parents/guardians and communities for their willingness to participate in this study. We gratefully acknowledge the assistance of staff at Albinama Health Centre and of the network of village-based health workers. We acknowledge the efforts of the PNG Institute of Medical Research Maprik field, administration, and laboratory staff, as well as technical support through other PNGIMR branches. We specially thank Anna Rosanas-Urgell and Alice Ura from PNGIMR for preserving RNA of samples.

## Author Contributions

**Conceptualization:** Leanne J. Robinson, Ivo Mueller, Ingrid Felger, Michael T. White.

**Data curation:** Aurel Holzschuh, Michael T. White.

**Formal analysis:** Michael T. White.

**Funding acquisition:** Leanne J. Robinson, Ivo Mueller, Ingrid Felger.

**Investigation:** Aurel Holzschuh, Benson Kiniboro, Leanne J. Robinson, Ivo Mueller.

**Methodology:** Aurel Holzschuh, Maria Gruenberg, Natalie E. Hofmann, Rahel Wampfler.

**Supervision:** Ingrid Felger.

**Visualization:** Michael T. White.

**Writing – original draft:** Aurel Holzschuh.

**Writing – review & editing:** Aurel Holzschuh, Ingrid Felger, Michael T. White.

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
