## [Decision Letter · Decision Letter 0]

18 Jun 2022

Dear Dr. White,

Thank you very much for submitting your manuscript "Co-infection of the four major Plasmodium species: effects on densities and gametocyte carriage" for consideration at PLOS Neglected Tropical Diseases. As with all papers reviewed by the journal, your manuscript was reviewed by members of the editorial board and by several independent reviewers. The reviewers appreciated the attention to an important topic. Based on the reviews, we are likely to accept this manuscript for publication, providing that you modify the manuscript according to the review recommendations. 

Please change P. ovale to P. ovale sp. - you do not differ between P. ovale curtisi and P. ovale wallikeri. 

Moreover the shorten forms for parasite species (like Pf) might not be in accordance to the authors guidelines - please check.

Sincerely,

Hans-Peter Fuehrer

Deputy Editor

Please change P. ovale to P. ovale sp. - you do not differ between P. ovale curtisi and P. ovale wallikeri. 

Moreover the shorten forms for parasite species (like Pf) might not be in accordance to the authors guidelines - please check.

Reviewer's Responses to Questions

**Key Review Criteria Required for Acceptance?**

**Methods**

-Are the objectives of the study clearly articulated with a clear testable hypothesis stated?

-Is the study design appropriate to address the stated objectives?

-Is the population clearly described and appropriate for the hypothesis being tested?

-Is the sample size sufficient to ensure adequate power to address the hypothesis being tested?

-Were correct statistical analysis used to support conclusions?

-Are there concerns about ethical or regulatory requirements being met?

Reviewer #1: This manuscript written by Holzschuh et al. investigates human Plasmodium species coinfections, its impact on gametocyte production and parasite densities. The manuscript is very clearly written, and well structured.

In detail: Objectives are clearly stated and the methods are well described. Statistical methods are well explained and adequate.

Overall: information given in the main text is rather short, but all of the information can be found in the supplement. I recommend to add more information to the main text, so that the reader does only have to go for the supplement for very detailed information (short information on villages, on sample collection, i.e. filter paper, volume of blood used).

Reviewer #2: The study is clearly articulated and the study design is appropriate with also clear hypothesis.

the sample size is also well sufficient.

Reviewer #3: Methods are sound. There are no concerns about the methods or ethical concerns; the data is thoroughly analyzed and interpreted with caution. The major achievement of the authors, developing sensitive methodologies for gametocyte carriage of all four species is an important one and makes this article of great value.

**Results**

-Does the analysis presented match the analysis plan?

-Are the results clearly and completely presented?

-Are the figures (Tables, Images) of sufficient quality for clarity?

Reviewer #1: The results are clearly presented and the figures are of sufficient quality.

One point is confusing: There are 505 pre-treatment samples mentioned, but according to methods only 504 children are included. Please explain the discrepancy.

Many results are only presented in the supplement. If this is not a requirement I suggest to add more results to the main text, e.g. the tables stating the positivity for Plasmodium species (asexual/sexual) for the different analysis

Reviewer #2: The results are missing a clear description or statment of some data: the Plasmodium species blood-stage infections, and Pf and Pv gametocytes (see general comments)

Reviewer #3: Figures 1-3 are clear and highly valuable. The Venn diagram nicely illustrates the richness (and complexity) of the data. Figure 4 is nice but somewhat misleading since all effects, even those that are not statistically significant, are depicted. It is unclear what the grey scare for Po gametocyte density vs Pv asexual density means. I would suggest to only present colors for those associations approaching statistical significance. 0.05 is obviously not sacred and it would be acceptable to only those associations with, for example, p<0.1. Now the light green suggests, for example, a increased density but it is never statistically significant and the effect size is very small. That may lead to wrong interpretations. Alternatively, the number of observations on which the effect size is based may be presented in the cells.

Figure 5 should include number of observations.

**Conclusions**

-Are the conclusions supported by the data presented?

-Are the limitations of analysis clearly described?

-Do the authors discuss how these data can be helpful to advance our understanding of the topic under study?

-Is public health relevance addressed?

Reviewer #1: The conclusion section is also well written and is supported by the data. Limitations of the study are described. One additional limitation that should be mentioned is the paucity of positive samples for P. ovale, so that it is difficult to draw conclusions. (This might go too far – but - In addition, as it is now widely accepted in the community that P. o wallikeri and curtisi are two distinct species, that are sympatric and not recombining, and therefore one might have to consider them separately, then it is even less samples).

Reviewer #2: The study general conclusion is supported by the data presented but the limitations should be improved with the specification of the co-infections densities and Pv and Pf gametocytes. The age of the samples could also be limiting for an actual application for reflexion of the results.

Reviewer #3: The discussion is balanced. Authors do not over-interpret their findings.

**Editorial and Data Presentation Modifications?**

Reviewer #1: accept

Reviewer #2: see below in summary and general comments.

Reviewer #3: None

**Summary and General Comments**

Reviewer #1: This manuscript presents very well conducted and written work on human Plasmodium species coinfections, its impact on gametocyte production and parasite densities. A novel PCR for detection of Po and Pm gametocytes has been established and is presented. I recommend to publish the manuscript.

Reviewer #2: The paper is presenting data on “Co-infection of the four major Plasmodium species: effects on densities and gametocyte carriage”. As described in the methods section, 504-505 children aged 5-10 years were actively monitored for infection by qPCR and illness for 32 weeks and blood samples were collected every 2 weeks for the first 12 weeks and every 4 week 14-32. While the topic and results are very interesting, there is an important lack of information of the data used, and also discussion of the results using the sampling structure.

These are my comments:

It is not clearly specified in the paper if the Plasmodium species blood-stage infections, and Pf and Pv gametocytes were analysed one more time together with Po and Pm gametocytes using the same RNA extraction. As the study is using more than 10 years old samples, and the same samples being used in different publications, the authors should clearly specify the exact data newly done in the current study and those taken from old studies. 

The information in methods section and supplementary data are showing that only Pm and Po gametocytes study were newly done. The authors need to clarify if they extracted again the RNA or they used left over RNA from previous studies. 

In case all analysis were repeated again with newly extracted RNA, the obtained co-infections parasitemia data and Pv, Pf gametocytes data should be discussed with the old data from the same authors.

My second main comment is why the authors did not discuss the data using the sampling structure: a 32 weeks retrospective sampling was done with 9 to 10 sample collection per child. It would have been very interesting to analyse and discuss the co-infections pattern of the same sample source during the 32 weeks.

The obtained co-infections of the 4 plasmodium species in PNG are for more than 10 years again. These data should be discussed using data on plasmodium evolution in PNG.

Line153: please check the sample number. 505 is used in the results

Reviewer #3: Appropriate.

PLOS authors have the option to publish the peer review history of their article (what does this mean?). If published, this will include your full peer review and any attached files.

Reviewer #1: No

Reviewer #2: No

Reviewer #3: No

Figure Files:

Data Requirements:

Reproducibility:

References

---

## [Editor Report · Decision Letter 1]

8 Jul 2022

Dear Dr. White,

Thank you very much for submitting your manuscript "Co-infection of the four major Plasmodium species: effects on densities and gametocyte carriage" for consideration at PLOS Neglected Tropical Diseases. As with all papers reviewed by the journal, your manuscript was reviewed by members of the editorial board and by several independent reviewers. The reviewers appreciated the attention to an important topic. Based on the reviews, we are likely to accept this manuscript for publication, providing that you modify the manuscript according to the review recommendations. 

Dear Authors,

The response to the reviewers comments is missing. I only see the one for the journal editor.

BW

Hans-Peter Fuehrer

Sincerely,

Hans-Peter Fuehrer

Deputy Editor

Hans-Peter Fuehrer

Deputy Editor

Dear Authors,

The response to the reviewers comments is missing. I only see the one for the journal editor.

BW

Hans-Peter Fuehrer

Figure Files:

Data Requirements:

Reproducibility:

References

---

## [Editor Report · Decision Letter 2]

22 Aug 2022

Dear Dr. White,

We are pleased to inform you that your manuscript 'Co-infection of the four major Plasmodium species: effects on densities and gametocyte carriage' has been provisionally accepted for publication in PLOS Neglected Tropical Diseases.

Best regards,

Hans-Peter Fuehrer

Section Editor

---

## [Editor Report · Acceptance letter]

8 Sep 2022

Dear Dr. White,

We are delighted to inform you that your manuscript, "Co-infection of the four major Plasmodium species: effects on densities and gametocyte carriage," has been formally accepted for publication in PLOS Neglected Tropical Diseases.

Best regards,

Shaden Kamhawi

co-Editor-in-Chief

Paul Brindley

co-Editor-in-Chief
